# Usefulness of Intraductal Placement of a Dumbbell-Shaped Fully Covered Self-Expandable Metal Stent for Post-Cholecystectomy Bile Leaks

**DOI:** 10.3390/jcm12206530

**Published:** 2023-10-14

**Authors:** Keito Nakagawa, Saburo Matsubara, Kentaro Suda, Takeshi Otsuka, Masashi Oka, Sumiko Nagoshi

**Affiliations:** 1Department of Gastroenterology, Kumagaya General Hospital, Saitama 360-8567, Japan; kate-ill@hotmail.co.jp; 2Department of Gastroenterology and Hepatology, Saitama Medical Center, Saitama Medical University, Saitama 350-8550, Japan; leclearlshelly@gmail.com (K.S.); ohitoyosinokaze@yahoo.co.jp (T.O.); oka@dd.iij4u.or.jp (M.O.); snagoshi@saitama-med.ac.jp (S.N.)

**Keywords:** ERCP, post-cholecystectomy bile leak, FCSEMS

## Abstract

**Background and aims**: In the treatment of post-cholecystectomy bile leaks, endoscopic naso-biliary drainage (ENBD) or biliary stenting using plastic stents is the standard of care. Fully covered self-expandable metal stent (FCSEMS) placement across the sphincter of Oddi is considered a salvage therapy for refractory cases, but pancreatitis and migration are the major concerns. Intraductal placement of a dumbbell-shaped FCSEMS (D-SEMS) could avoid these drawbacks of FCMSESs. In this retrospective study, we investigated the usefulness of intraductal placement of the D-SEMS for post-cholecystectomy bile leaks. **Methods**: Six patients who underwent intraductal placement of the D-SEMS for post-cholecystectomy bile leaks were enrolled. This method was performed as initial treatment in three patients and as salvage treatment in three ENBD refractory cases. **Results**: Technical and clinical successes were obtained in 6 (100%) patients and 5 (83%) patients, respectively. One clinically unsuccessful patient required laparoscopic peritoneal lavage. The early adverse event was one case of mild pancreatitis (17%). The median duration of the D-SEMS indwelling was 61 days (42–606 days) with no migration cases, all of which were successfully removed. The median follow-up after index ERCP was 761 (range: 161–1392) days with no cases of recurrent bile leaks. **Conclusions**: Intraductal placement of the D-SEMS for post-cholecystectomy bile leaks might be safe and effective even in refractory cases.

## 1. Introduction

Bile leaks occur in 0.8–1.4% of patients undergoing laparoscopic cholecystectomy [1] and can lead to serious complications such as biliary peritonitis, abscess formation, and bleeding. Traditionally, surgical or percutaneous interventions were required; however, minimally invasive treatment with endoscopic retrograde cholangiopancreatography (ERCP) emerged several decades ago and is now considered the standard of care [2]. Various endoscopic techniques have been reported to treat bile leaks, including endoscopic sphincterotomy (EST) alone, biliary stenting with or without EST, and endoscopic naso-biliary drainage (ENBD). The advantage of EST alone is completion of the treatment in a single endoscopic procedure, with a reported treatment success rate of 88–91% [3,4]. However, a study comparing EST alone with biliary stenting in a canine model reported a significantly shorter time to healing in the stenting group [5]. Therefore, EST alone is considered an option in minor leaks [6,7]. Biliary stenting using a plastic stent (PS) with or without EST is widely performed, with reported success rates of 53–100% [4,8,9,10,11,12,13,14,15,16,17]; some cases are refractory and may require additional treatments including percutaneous or surgical interventions [17,18]. ENBD has several advantages such as reliable biliary decompression due to the absence of duodeno-biliary reflux, repeat cholangiography to confirm healing of the bile leak, and a single endoscopic session. However, patients’ discomfort and long hospital stay are major drawbacks of ENBD. Biliary stenting using a fully covered self-expandable metal stent (FCSEMS) has emerged as a salvage method for cases refractory to PS or ENBD, with reported success rates of almost 100% [19,20,21,22,23,24,25]. However, post-ERCP pancreatitis and stent migration are major concerns of FCSEMS placement.

Recently, a dumbbell-shaped FCSEMS (D-SEMS) (BONA stent M-intraductal; Standard Sci Tech, Seoul, Republic of Korea) (Figure 1) has been developed for the treatment of benign biliary strictures [26]. This FCSEMS is designed for intraductal placement with a short length (30/40/50/60/70 mm), a long lasso on the distal end for removal, and a central waist for prevention of migration, and it has rounded-shape at both ends to minimize stent-induced bile duct injury. Therefore, intraductal placement of the D-SEMS may fit the treatment of post-cholecystectomy bile leaks. In this study, we investigated the usefulness of intraductal placement of the D-SEMS for the treatment of post-cholecystectomy bile leaks.

## 2. Materials and Methods

### 2.1. Patients

This was a single-center, retrospective cohort study using prospectively collected ERCP data at Saitama Medical Center, Saitama Medical University. Between April 2017 and March 2022, consecutive patients who underwent ERCP with the D-SEMS for post-cholecystectomy bile leaks were enrolled. The post-operative bile leak was defined by the International Study Group of Liver Surgery (ISGLS) as an increased bilirubin concentration in the drain or ascites at least 3 times greater than the serum bilirubin concentration on or after post-operative day 3, or as the need for additional endoscopic or radiological interventions [27]. This study was conducted in accordance with the Declaration of Helsinki and approved by the Institutional Review Board of Saitama Medical Center, Saitama Medical University (ethical approval number 2022–101, date of approval: 12 January 2023). Written informed consent for ERCP was obtained from all patients, whereas informed consent for this study was waived due to opting out.

### 2.2. Procedures

ERCP was performed using a duodenoscope (TJF-Q290V, TJF-260V, or JF-260V; Olympus Medical Systems, Tokyo, Japan). Patients were sedated in the prone position with midazolam and pethidine hydrochloride. After selective biliary cannulation using wire-guided cannulation method, the location of the bile leak was detected via cholangiogram. Following the guidewire placement across the leak site, the D-SEMS was placed covering the leak site with the distal end above the sphincter of Oddi without EST (Figure 2). Diameter and length of the D-SEMS were selected at the discretion of the endoscopist.

### 2.3. Follow-Up

After successful placement of the D-SEMS, surgical drains were removed when output ceased. A second ERCP was performed two or three months after the index ERCP. Following removal of the indwelled D-SEMS through the accessory channel by pulling the lasso on the distal end using biopsy forceps, biliary canulation and contrast injection were performed to confirm the healing of the bile leak. After removal of the D-SEMS, patients were followed at the outpatient clinic to check the recurrence of symptoms.

### 2.4. Outcome Measures and Definitions

Outcome measures of this study were technical success, clinical success, early (up to 14 days) and late adverse events, and recurrent symptoms. Technical success was defined as the appropriate placement of the D-SEMS as intended. Clinical success was defined as the removal of surgical drains without additional surgical or percutaneous interventions, with no recurrent symptoms after the removal of the D-SEMS [25]. Adverse events were described in accordance with the American Society for GI Endoscopy lexicon [28]. The severity of the bile leak was classified into two groups: low-grade (leak visualized only after intrahepatic bile duct opacification) or high-grade (leak depicted before intrahepatic bile duct opacification) [4].

### 2.5. Statistics

Continuous variables and categorical variables are presented as median (range) and number (percentage), respectively. All statistical analyses were performed using JMP Pro version 16.0.0 (SAS, Cary, NC, USA).

## 3. Results

### 3.1. Flowchart and Characteristics of the Study Patients

In the initial phase of the study period (through September 2019), ENBD was performed as the initial treatment, with a conversion to intraductal D-SEMS placement for refractory cases. In the latter phase, we performed intraductal D-SEMS placement as the initial treatment. During the entire period, five low-grade leak cases were successfully treated using ENBD (N = 2) or D-SEMS (N = 3), and three high-grade leak cases were treated using ENBD but failed and subsequently converted to intraductal D-SEMS placement. In total, six patients underwent intraductal placement of the D-SEMS (Figure 3).

Characteristics of the study patients are shown in Table 1. The primary disease requiring cholecystectomy was acute cholecystitis in 3 (50%) patients and gallbladder stones in 3 (50%) patients. The type of surgery was laparoscopic in 4 (67%) patients and open in 2 (33%) patients. The leak site was the cystic duct stump in 5 (83%) patients and common bile duct in 1 (17%) patient. The severity of bile leak was low-grade in 3 (50%) patients and high-grade in 3 (50%) patients. Intraductal D-SEMS placement was performed as a primary intervention in 3 (50%) patients and as salvage therapy for refractory to ENBD in 3 (50%) patients. All ENBD refractory cases were high-grade leaks, including one case of leakage from the common bile duct.

In the three cases converted from ENBD, the duration of the ENBD tube indwelling was 7, 17, and 23 days. The median time from surgery to ERCP (for D-SEMS placement) was 10 days (range: 3–34 days).

### 3.2. Outcomes

Procedure details and study outcomes are described in Table 2. Technical and clinical successes were obtained in 6 (100%) patients and 5 (83%) patients, respectively. One clinically unsuccessful patient underwent laparoscopic peritoneal lavage the day after the procedure. This was a case of a high-grade leak refractory to ENBD, leading to abscess formation.

The median time to the clinical success was 5 days (range: 3–24 days). The early adverse event was one case of mild pancreatitis (17%). The median duration of the D-SEMS indwelling was 61 days (42–606 days) with no migration cases, all of which were successfully removed. There were no cases of sustained bile leaks on the cholangiogram. One patient temporarily dropped out from the outpatient clinic and retained the D-SEMS for 606 days, resulting in the stent–stone complex. The D-SEMS was removed after crushing the stones by electrohydraulic lithotripsy using a per-oral cholangioscope. The median follow-up after index ERCP was 761 (range: 161–1392) days with no cases of recurrent bile leaks.

## 4. Discussion

In the current study, intraductal placement of the D-SEMS resolved post-cholecystectomy bile leaks in 5 of 6 cases (83%), including ENBD-refractory cases. One failed patient required laparoscopic peritoneal lavage, but the subsequent course was uneventful, indicating successful sealing of the bile leak with the D-SEMS. If conversion from ENBD had been performed earlier, surgery might not have been necessary.

The traditional concept of endoscopic treatment for bile leaks is to disable the sphincter of Oddi and to reduce the pressure in the bile duct [29] using EST, PS placement, and ENBD. In addition, PS placement and ENBD allow bile to bypass the leak site when bridging stenting is possible, facilitating closure of the leak site [6,30]. Several previous studies have reported that PS placement is more effective than EST alone in healing the leak site [5,30]. However, since PSs and an ENBD tube cannot directly seal the leak site, it takes time until leaks cease and sometimes the leak site cannot be healed in high-grade leak cases.

FCSEMSs are thick and self-expanding, and their ability to adhere to the bile duct wall allows them to directly seal the leak site and immediately stop the leakage if bridging is achieved. As a result, healing of the leak site is also easier to obtain. In fact, in a study comparing multiple PSs and an FCSEMS in patients refractory to single PS placement, the latter was superior [24]. Also, one report showed that patients refractory to multiple PSs were successfully treated with an FCSEMS [13]. These facts indicate the importance of direct sealing of the leak site.

To date, conventional tubular-type FCSEMSs have been placed across the sphincter of Oddi for the treatment of bile leaks. As a result, pancreatitis, migration, sludge/gallstone formation due to reflux of duodenal fluid, and bile duct injury due to the proximal end of the stent have been problematic [25,29,31]. The use of FCSEMSs in cases other than pancreatic cancer or chronic pancreatitis is a known risk factor for pancreatitis [32]. EST should be performed to prevent pancreatitis, but the preventive effect is not sufficient [33], and there is a risk of bleeding or perforation due to EST [16]. Since bile leak cases do not have biliary strictures, FCSEMSs are likely to migrate. Although there are some reports of the use of FCSEMSs with anchoring fins [20] or partially covered SEMS [31] to prevent migration, there are also reports of unremovable cases.

Intraductal placement of the D-SEMS is a treatment method that specializes in directly sealing the leak site, rather than in reducing the biliary pressure. In the present study, clinical success was as good as that of the conventional FCSEMS placement across the sphincter of Oddi and had the following advantages. First, the incidence of pancreatitis was low despite the absence of EST, because the pancreatic orifice was not blocked. Second, despite the absence of strictures, no migration was observed. This may be due to the dumbbell shape and the lack of influence of peristalsis of the duodenum. Third, no stent-induced bile duct wall injury/stricture was observed, and all stents were able to be removed, including a case with the D-SEMS indwelling for 606 days. The rounded ends minimize damage to the bile duct wall, and the distal end is retracted when the lasso is pulled, allowing easy removal through the channel. There has been one report on intraductal placement of the D-SEMS for post-operative bile leaks, with a low success rate of 5/7 (71%) [34]. However, it should be considered that difficult cases such as partial hepatectomy and living donor liver transplantation were included. To the best of our knowledge, the present study is the first report of intraductal placement of the D-SEMS limited to post-cholecystectomy bile leaks, showing leak cure in all of six cases.

When should the D-SEMS be used? Commonly, the indication for FCSEMSs for bile leaks is considered PS-refractory cases [30]. The risks of PS failure are reported as high-grade leak [13] or leakage from sites other than the cystic duct stump such as the common bile duct or intrahepatic bile ducts [15,17,35,36]. In the current study, all patients with high-grade leak were refractory to ENBD and converted to the D-SEMS, while all patients with low-grade leak were successful with ENBD or the D-SEMS. One patient with leakage from the common bile duct was refractory to ENBD. These results may validate previous reports. Since failure of PS placement or ENBD may lead to surgical procedures as seen in the present study, the D-SEMS can be suitable as the initial therapy for high-grade leaks or leaks from sites other than the cystic duct stump. Although the D-SEMS is routinely not necessary for low-grade leaks considering the cost, it is acceptable to initially use the D-SEMS in patients with bleeding tendency or in elderly patients who are hesitant to undergo multiple endoscopic procedures, because EST is not necessary and it provides more reliable closure of the leak site than PSs.

There were several limitations in the present study. First, it was a retrospectively designed single-center study. Therefore, selection bias and reporting bias could not be eliminated. Second, only a small number of patients were included. Third, patients who underwent PS placement, the standard treatment for post-operative bile leaks, were not included.

In conclusion, intraductal placement of the D-SEMS for post-cholecystectomy bile leaks might be safe and effective even in cases refractory to ENBD. It can be a promising treatment option for post-cholecystectomy bile leaks, especially for the high-grade leak or the leak from sites other than the cystic duct stump. Further investigations with a prospective design in a large number of patients are warranted.

## Figures and Tables

**Figure 1 jcm-12-06530-f001:**
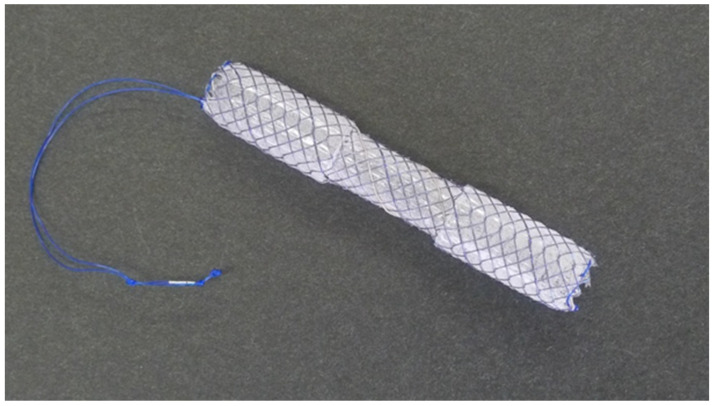
D-SEMS: a dumbbell-shaped fully covered self-expandable metal stent (BONA stent M-intraductal; Standard Sci. Tech., Seoul, Republic of Korea) (courtesy of Medico’s Hirata, Osaka, Japan).

**Figure 2 jcm-12-06530-f002:**
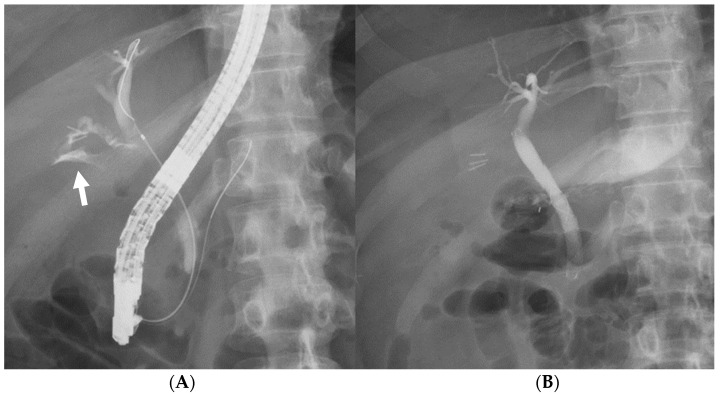
D-SEMS placement. (**A**) The bile leak was depicted via endoscopic retrograde cholangiopancreatography at the cystic duct stump (white arrow). (**B**) An 8 mm × 50 mm D-SEMS was placed covering the cystic duct take-off above the sphincter of Oddi.

**Figure 3 jcm-12-06530-f003:**
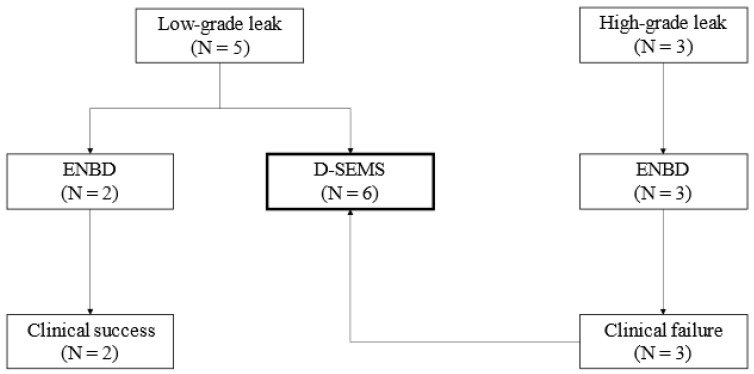
Flow chart of the study patients. ENBD, endoscopic naso-biliary drainage; D-SEMS, dumbbell-shaped fully covered self-expandable metal stent.

**Table 1 jcm-12-06530-t001:** Baseline characteristics of the study patients.

Age	73.5 (51–87)
Sex, male	2 (33)
ASA-PS	2.5 (2–4)
CCI	6 (2–7)
Primary disease, AC/GBS	3 (50)/3 (50)
Type of surgery, laparoscopic/open	4 (67)/2 (33)
Site of leak, cystic duct stump/common bile duct	5 (83)/1 (17)
Severity of leak, low-grade/high-grade	3 (50)/3 (50)
Timing of intervention, primary/conversion from ENBD	3 (50)/3 (50)
Time from surgery to ERCP, days	10 (3–34)

Numbers are shown in number (%) or median (range). ASA-PS, American Society of Anesthesiologists physical status; CCI, Charlson comorbidity index; AC, acute cholecystitis; GBS, gallbladder stones; ENBD, endoscopic naso-biliary drainage; ERCP, endoscopic retrograde cholangiopancreatography.

**Table 2 jcm-12-06530-t002:** Procedure details and study outcomes.

Length of D-SEMS, 50 mm/60 mm	4 (67)/2 (33)
Diameter of D-SEMS, 10 mm/8 mm	5 (83)/1 (17)
Technical success	6 (100)
Clinical success	5 (83)
Time to the clinical success, days	5 (3–24)
Early adverse events (mild pancreatitis)	1 (17)
Late adverse events (stent–stone complex)	1 (17)
D-SEMS indwelling period, days	61 (42–606)
Follow-up period after index ERCP, days	761 (161–1392)

Numbers are shown in number (%) or median (range). D-SEMS, dumbbell-shaped fully covered self-expandable metal stent; ERCP, endoscopic retrograde cholangiopancreatography.

## Data Availability

The data generated during this study are available within the article. Datasets analyzed during the current study preparation are available from the corresponding author on reasonable request.

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
