# Peer review of "Usefulness of Intraductal Placement of a Dumbbell-Shaped Fully Covered Self-Expandable Metal Stent for Post-Cholecystectomy Bile Leaks"

_jcm, 2023, doi:10.3390/jcm12206530_

Round 1
Reviewer 1 Report
It is a interesting study to compare M-intraductal self-expandable metal stents(SEMSs) with ENBD for bile leakage after operation. However, the case number is to small for compare the difference between two groups. The SEMSs seems to be good rescue tool after ENBD failure for the bile leakage. I think the authors may put more emphasizing about the efficacy for the M-intraductal SEMS
The quality of English is fine.
Author Response
Thank you for your valuable comment.
Please see the attachment.
We would like to change the manuscript type from "article" to "communication" because of small sample size and short length of the main text.

Reviewer 2 Report
Dear authors
I kindly review your manuscript and unfortunately I encountered several major concerns. Firstly and most importantly your paper provide a comparative analysis basically of 2 endoscopic approaches for biliary leaks following cholecistectomy. The sample size, in my opinion is to low to statistically justify the feasibility of a possible comparison, merely allowing an aneddotical description of a case series. Consequently the use of inferential statistic means such as Mann-Whitney test or Fisher exact test is not accurate. It's impossible to clearly state a superiority of a technique over another from the description of 8 cases. Moreover it seems that one of the main characteristics relevant for comparison is the leak details. The issue is that the definition of low and high grade is far from clear and all the 3 cases treated first-line with D-SEMS are classified as low-grade versus 3 high-grade from the ENBD group, suggesting a real difference in the severity of leaks from one group over the other, impacting the results. I suggest to enlist more cases and to design for example a case-control matching at enrollment study (if a porspective evaluation is impossible). It should improve your results strenghtening your hypotesis. Finally extensive english revisions are warranted
Kind regards
Extensive english revisions and formatting mistakes
Author Response

(The authors gave the same response as above.)

Reviewer 3 Report
The authors' article is potentially of interest to hepatobiliary clinicians and contributes to broadening the corpora of clinical evidence for the management of biliary leaks.
The authors conclude that D-SEMS may be the first-line treatment for post-cholecystectomy bile leaks; however, I am not sure if this conclusion can be made based on a sample size of 8. Would you please provide your rationale for this conclusion.
Regards.
The English quality of the manuscript is overall good. There may be a couple typos here and there that may need fixing, otherwise the manuscript is fine.
Author Response

(The authors gave the same response as above.)

Round 2
Reviewer 1 Report
The result may provide good option for endoscopist after ENBD or plastic stent failure to treat bile leakage. Further study to increase the number of the M-intraductal SEMS is demanded
Author Response
Thank you for your kind and valuable comment.
Reviewer 2 Report
Dear authors
I kindly reviewed your revised version of the paper and i gladly noticed the adjustments requested in order to present it as a case series. Also english editing and format improvements (including tables) ameliorated the overall presentation of your paper. I suggest only minor english revision but your case series now has the potential to advocate interest towards D-SEMS placement in post-cholecistectomy bile leaks. I encourage you to design a prospective mono or multicenter study to put statistics and methodological rigor into this comparison with ENBD.
Minor revisions
Author Response
Thank you for your kind and various comments. We have revised in english manner throughout the manuscript.